# Current Therapeutic Options and Potential of Mesenchymal Stem Cell Therapy for Alcoholic Liver Disease

**DOI:** 10.3390/cells12010022

**Published:** 2022-12-21

**Authors:** Jinsol Han, Chanbin Lee, Jin Hur, Youngmi Jung

**Affiliations:** 1Department of Integrated Biological Science, College of Natural Science, Pusan National University, Pusan 46241, Republic of Korea; 2Institute of Systems Biology, College of Natural Science, Pusan National University, Pusan 46241, Republic of Korea; 3Department of Convergence Medicine, Pusan National University School of Medicine, Yangsan 50612, Republic of Korea; 4PNU GRAND Convergence Medical Science Education Research Center, Pusan National University School of Medicine, Yangsan 50612, Republic of Korea; 5Department of Biological Sciences, College of Natural Science, Pusan National University, Pusan 46241, Republic of Korea

**Keywords:** alcoholic liver disease, therapeutics, mesenchymal stem cells, cell-free therapy

## Abstract

Alcoholic liver disease (ALD) is a globally prevalent chronic liver disease caused by chronic or binge consumption of alcohol. The therapeutic efficiency of current therapies for ALD is limited, and there is no FDA-approved therapy for ALD at present. Various strategies targeting pathogenic events in the progression of ALD are being investigated in preclinical and clinical trials. Recently, mesenchymal stem cells (MSCs) have emerged as a promising candidate for ALD treatment and have been tested in several clinical trials. MSC-released factors have captured attention, as they have the same therapeutic function as MSCs. Herein, we focus on current therapeutic options, recently proposed strategies, and their limitations in ALD treatment. Also, we review the therapeutic effects of MSCs and those of MSC-related secretory factors on ALD. Although accumulating evidence suggests the therapeutic potential of MSCs and related factors in ALD, the mechanisms underlying their actions in ALD have not been well studied. Further investigations of the detailed mechanisms underlying the therapeutic role of MSCs in ALD are required to expand MSC therapies to clinical applications. This review provides information on current or possible treatments for ALD and contributes to our understanding of the development of effective and safe treatments for ALD.

## 1. Introduction

Alcohol has long been recognized as a critical risk factor for many diseases [1,2]. However, alcohol consumption is not well controlled because of its addictive properties and social or cultural needs [3,4]. Uncontrolled, chronic and binge alcohol consumption result in an increase in alcohol-related diseases worldwide and account for 5.1% of the global burden of diseases [5]. Alcoholic liver disease (ALD) is responsible for the majority of alcohol-related deaths [6,7]. The liver is highly susceptible to alcohol because it is the first organ where alcohol is metabolized, and it has a high level of alcohol-metabolizing enzymes [8,9]. Metabolization of alcohol in the liver produces various hepatotoxic byproducts and significant oxidative stress on the liver, leading to the large-scale death of hepatocytes [8,10]. Oxidative stress and excessive cell death exacerbate inflammation in the liver [8,11]. Prolonged cell damage and inflammation activate hepatic stellate cells (HSCs), which are key players in the development of fibrosis in the liver [8,12]. ALD encompasses a diverse spectrum, from mild to severe pathologies, including steatosis, steatohepatitis, cirrhosis, and hepatocellular carcinoma (HCC) [13]. Given the prevalence of ALD worldwide, concerted efforts have been made to control alcohol consumption [11,14]. However, the trend in alcohol consumption is steadily increasing [13]. The COVID-19 pandemic has accelerated the prevalence of ALD [15]. There are no Food and Drug Administration (FDA)-approved drugs specifically targeting ALD, and the only treatments for ALD are abstinence and liver transplantation [16]. Thus, there is an urgent need for the development of ALD therapeutics. Several drugs are currently prescribed to patients with ALD as a supportive measure to delay death or to maintain health until liver transplantation is possible [17,18,19]. However, these available options are insufficient and/or ineffective for the patients [17,18,19]. 

In the absence of effective drug treatments, researchers are focusing on therapeutic strategies, targeting ALD pathogenesis and oxidative stress, regeneration, and inflammation [18,20,21]. Stem cell therapy has emerged as a promising therapy for ALD based on the immunomodulatory and regenerative capacities of stem cells in liver diseases, including ALD [22,23,24]. Among various types of stem cells, mesenchymal stem cells (MSCs) are considered a strong candidate for stem cell therapies because they are multipotent and can be obtained relatively easily from various sources, such as bone marrow (BM), adipose tissue, placenta, and umbilical cord (UC) [25,26,27]. In addition, MSCs do not give rise to the same ethical controversy that embryonic stem cells do [27,28,29]. MSCs have been widely studied and tested in clinical trials of liver diseases [22,23,24]. The therapeutic potential of MSC-derived secretory factors for liver diseases has been proven [27,30]. Herein, we briefly discuss the pathogenesis of ALD and summarize current therapeutic options for ALD and their limitations. We also review possible therapeutic strategies for ALD based on MSCs.

## 2. Pathogenesis of ALD

After alcohol is transferred through the bloodstream into the liver, hepatic enzymes convert the alcohol into acetaldehyde [8,9,31,32]. Among hepatic enzymes, alcohol dehydrogenase (ADH) and cytochrome P450 2E1 (CYP2E1) are the two main enzymes that oxidize alcohol to acetaldehyde, which plays a major role in alcohol-induced hepatotoxicity [8,9]. Acetaldehyde is then further metabolized to acetate by aldehyde dehydrogenase (ALDH) [8,9]. Most acetate leaves the liver and is metabolized into carbon dioxide, fatty acids, and water in peripheral tissues [8,9,10]. When an excessive amount of alcohol is ingested, the expression and activity of CYP2E1 rather than ADH is enhanced, and CYP2E1 converts alcohol to acetaldehyde, resulting in the production of reactive oxygen species (ROS) such as superoxide, peroxynitrite, hydrogen peroxide, and hydroxyl radicals [8,9,10,33,34]. The role of ROS in promoting oxidative stress is well known [35,36,37]. Prolonged alcohol consumption impairs liver lipid metabolism and leads to excessive hepatic fat accumulation by increasing fatty acid uptake and de novo lipogenesis and decreasing β-oxidation and secretion of very low-density lipoproteins [8,38,39]. This results in the accumulation of massive hepatic lipids in hepatocytes producing ROS [40,41]. Excessive ROS promotes lipid peroxidation and generates malondialdehyde (MDA) and 4-hydroxynonenal, which form toxic proteins or DNA adducts with acetaldehyde [42,43]. Alcohol also disrupts the antioxidant defense system by lowering the levels of antioxidants, including glutathione (GSH) and S-adenosyl-L-methionine (SAMe), provoking oxidative stress [44]. Oxidative stress impairs mitochondrial function by inducing abnormal enlargement of mitochondria and mitochondrial DNA damage and reducing hepatic ATP levels and mitochondrial protein synthesis [42,45,46,47]. In addition, excessive alcohol injures the successful repair process by hepatocytes and progenitors [48,49]. Hepatocytes are known to possess regenerative capacity to refill the loss of liver mass in response to liver damage [50,51]. Hepatic injury induces the upregulation of DNA synthesis in remaining mature hepatocytes and/or triggers expansion of the progenitor cell population, with these cells differentiating into hepatocytes [50,51]. As alcohol significantly inhibits the proliferation of both mature hepatocytes and liver progenitor cells and interrupts the differentiation of liver progenitors, immature and nonfunctional hepatocytes accumulate in the liver [49,52,53]. 

Dying hepatocytes damaged by alcohol release various cytokines and chemokines, such as tumor necrosis factor-α (TNF-α), interleukin (IL)-6, and monocyte chemoattractant protein-1 (MCP-1) [54,55]. These activate liver-resident macrophages known as Kupffer cells to recruit neutrophils and monocytes into the liver [54,55,56]. These inflammatory cells produce a wide variety of cytokines, which activate multiple signaling pathways in the liver. Among these pathways, the roles of nuclear factor-κB (NF-κB) and signal transducer and the activator of transcription 3 (STAT3) in the pathogenesis of liver diseases have been extensively studied [54,56,57,58]. NF-κB activated by alcohol metabolites induces the expression of various genes encoding pro-inflammatory cytokines and chemokines and participates in inflammasome regulation [58]. In response to alcohol injury, IL-6 is released from Kupffer cells and activates STAT3 in hepatocytes. Activated STAT3 promotes the production of pro-inflammatory cytokines and chemokines in these cells and increases monocyte/macrophage infiltration into the liver, exacerbating inflammation [57]. Hepatic inflammation is accompanied by fibrosis [59,60]. Liver fibrosis impairs hepatic function and architecture, leading to death from liver failure [61,62]. HSCs are a key contributor to fibrogenesis [63,64]. In damaged liver, HSCs gradually lose their distinctive features and undergo trans-differentiation into myofibroblast-like HSCs in a process called activation [63,64]. Profibrotic factors stimulating HSC activation, such as transforming growth factor-β (TGF-β), Hedgehog, and platelet-derived growth factor, are reported to be upregulated in patients with ALD and in animal models of ALD [65,66,67] (Figure 1). 

## 3. Current Therapies and New Targets for ALD

Efforts to overcome ALD have been made for a long time, and many therapeutic agents and approaches have been tested in experimental animal models of ALD and even in ALD patients [17,18,19,20,21]. To date, however, no drug has been approved by the FDA [16]. Hence, understanding the mechanisms and limitations of the effects of therapeutic options currently being tested and applied could provide prospections for the discovery of novel therapeutic agents for ALD. This section summarizes the current available and newly proposed treatments for ALD and reviews the mechanisms underlying their effects on the disease (Table 1).

### 3.1. Management of Alcohol Abuse

It is vital to emphasize the importance of abstinence to patients with ALD [7,68,69]. Abstinence ameliorates liver damage caused by alcohol consumption and reverses liver damage in patients with early-stage ALD [69,70]. However, abstinence is the most difficult therapy, given that most patients with ALD have alcohol usage disorders (AUD), such as alcoholism [4,71]. Thus, several drugs, such as naltrexone, baclofen, and acamprosate, are prescribed to AUD patients to reduce their craving for alcohol [72,73]. The mechanism underlying the action of naltrexone in alcoholism is poorly understood [74]. According to preclinical data, naltrexone competitively binds to opioid receptors in the central nervous system and blocks the effects of endogenous opioids, reducing the desire for alcohol [75]. Both baclofen and acamprosate interfere with the action of the neurotransmitter γ-aminobutyric acid [76,77]. Disulfiram, another drug to manage ALD, causes aversion to alcohol by inhibiting the major alcohol metabolizing enzyme, ADH, and producing more acetaldehyde, resulting in severe hangovers and unpleasant feelings after consuming alcohol [78,79]. However, for ALD patients with advanced liver disease, pharmacological treatments are limited because impaired liver function affects drug metabolism and can increase the risk of drug-related hepatotoxicity [80,81]. Furthermore, despite abstinence, some cases of ALD can progress to cirrhosis [82,83]. In addition to managing alcohol consumption among patients with ALD, treatments alleviating the pathogenesis of ALD are needed. A number of studies have investigated treatments targeting specific pathways involved in the pathogenesis of ALD with the aim of finding novel therapeutic strategies or targets. 

### 3.2. Antioxidants Alleviating Oxidative Stress in ALD

Because oxidative stress plays an important role in alcohol-induced liver damage, various antioxidants have been used to mitigate severe oxidative stress and reduce hepatic injury in ALD [84,85]. Among antioxidants used for ALD, vitamin E, N-acetylcysteine (NAC), and SAMe are the most well-known and well-studied [84,85,86]. Vitamin E prevents lipid peroxidation in cellular membranes by donating an electron to neutralize free radicals [87]. Vitamin E has attracted attention in clinical trials because the level of vitamin E is frequently low in patients with alcoholic cirrhosis [88,89,90]. Several studies have proven the hepatoprotective role of vitamin E in animal models of ALD [91,92]. In these studies, vitamin E restored the redox status, prevented oxidative stress, and reduced apoptosis by inhibiting NF-κB activation, ameliorating liver damage caused by alcohol [92]. In addition, TNF-α, CYP2E1, TGF-β1, and collagen type I expression and liver fibrosis were alleviated significantly in an alcohol-administered group with vitamin E [91]. Raxofelast, an analog of vitamin E, showed a similar effect to that of vitamin E in a mouse model of ALD [93]. However, significant therapeutic effects were not observed in clinical trials, with neither liver function nor the 1-year survival rate improving in patients with ALD treated with vitamin E [94,95]. However, Miyashima et al. [96] reported that combination therapy with vitamin E and corticosteroids, which modulate inflammation, rapidly improved liver enzymes and coagulopathy in patients with severe alcoholic hepatitis (AH) and suggested that combination treatment could be a supportive therapeutic option for patients with advanced ALD. 

NAC, a precursor of intracellular cysteine and GSH, supplements the intracellular level of GSH, which is a fundamental antioxidant synthesized in cells and is involved in removing anti-inflammatory ROS [97]. As shown in several rat models, NAC reduces alcohol consumption and stabilizes the glutamate system, improving cortical glutamatergic neurotransmission damaged by alcohol [98,99,100]. Based on these beneficial effects of NAC, it has been tested in experimental animal models of, and in patients with, liver diseases, such as acute liver failure, nonalcoholic fatty liver disease (NAFLD), ALD, and drug-induced liver injury [101,102,103,104,105]. As GSH deficiency is a pathophysiological characteristic of ALD, NAC is thought to be a promising therapeutic agent for patients with ALD [106,107,108]. In preclinical studies using rodents, NAC treatment enhanced cytosolic antioxidant activity and inhibited lipid peroxidation [109]. The treatment also significantly ameliorated liver injury and inflammation with maintained hepatic GSH content in alcohol-fed rodents [109]. Despite these promising results in animal models, several clinical studies found that NAC did not have a therapeutic effect on patients with severe AH [110]. Administration of NAC combined with glucocorticoids or other antioxidants, such as vitamins A–E, zinc, copper, magnesium, folic acid, and coenzyme Q, improved short-term survival (i.e., 1–2 months) but not longer-term survival (i.e., more than 6 months) in patients with severe AH [111]. In line with these findings, Singh et al. [112] reported that combination therapy with NAC and granulocyte colony stimulating factor (G-CSF) hardly have benefits for patients with severe AH compared with G-CSF alone. In a randomized study, NAC administration to nine patients with alcohol use disorders had little impact on alcohol consumption and physiological outcomes [113]. 

SAMe is a major methyl donor in the transmethylation reaction regulating GSH synthesis [114]. Alcohol is known to significantly reduce the levels of both hepatic SAMe and mitochondrial GSH, a free radical scavenger, in both patients with ALD and animal models [115,116]. Treatment with SAMe has been shown to markedly reduce GSH depletion and increase plasma aspartate aminotransferase [117]. In baboons with ALD, replenishment of hepatic SAMe through oral administration attenuated ethanol-induced liver injury and improved mitochondrial dysfunction [118]. In addition, SAMe significantly alleviated the number of alcohol-induced megamitochondria in baboons with chronic alcohol consumption [118]. In patients with ALD, supplementing SAMe decreased mortality and delayed the need for a liver transplant, although this finding was not statistically significant [119]. Although many clinical trials of vitamin E and NAC for ALD have been conducted, there have been few clinical trials of SAMe. Of those that have been conducted, the findings have yet to be published. Hence, large, well-conducted, placebo-controlled randomized clinical trials are needed to obtain evidence for a therapeutic effect of SAMe in ALD [21,120,121]. In addition to vitamin E, NAC, and SAMe, various other antioxidants, such as curcumin, mitoquinone, oleanolic acid, and plant extracts, have been proposed for ALD therapy [122,123,124]. However, the therapeutic potential of these antioxidants is unclear [122,123,124]. 

### 3.3. Promoting Successful Liver Regeneration in ALD

Chronic alcohol consumption impairs the proliferation of hepatocytes and the differentiation of progenitors into hepatocytes [49,52,53,125]. Impaired liver regeneration results in the loss of liver function, with increased inflammation and fibrosis [126]. Hence, improving the regenerative activity of the liver could be a strategy to treat ALD. G-CSF, which is a glycoprotein, stimulates BM to produce granulocytes and stem cells and release these into the bloodstream [127]. In patients with alcohol-associated cirrhosis and steatohepatitis, the administration of G-CSF increased the number of clusters of differentiation 34^+^ cells, hematopoietic stem/progenitor cells, hepatic progenitors, and hepatocyte growth factor (HGF) levels in the liver, suggesting that G-CSF could enhance liver regeneration in patients with severe AH [128]. Improved survival of ALD patients who were treated with G-CSF and standard care with pentoxifylline—which regulates inflammation—has been reported versus that of patients treated with pentoxifylline only for 90 days [129]. Additional treatment with G-CSF after standard medical care with a corticosteroid for 30 days greatly decreased mortality among patients with ALD compared to patients treated only with standard medical care [130]. However, the mechanism explaining the treatment effect of G-CSF in ALD has not been elucidated, with clinical trials demonstrating the action of G-CSF accompanied by conventional therapies, not the specific effect of G-CSF alone on ALD [131]. 

Anti-inflammatory IL-22 influences liver regeneration. IL-22 is a member of the IL-10 cytokine family, which is secreted by various inflammatory cells, such as T helper 17, 22 and natural killer (NK) cells [132]. IL-22 has been shown to be associated with the maintenance of tissue homeostasis and repair by reducing oxidative stress, apoptosis, and steatosis in the liver [132,133]. In common with IL-10, STAT3 is the downstream signal transducer of IL-22 [132,133]. STAT3 is involved in mediating cell proliferation, survival, and differentiation [134]. The overexpression of *IL-22* in HepG2 cells, a human liver cancer cell line, constitutively activates STAT3 and induces the expression of antiapoptotic proteins, including B-cell lymphoma (Bcl)-2, Bcl-xL, and myeloid cell leukemia-1, and mitogenic proteins, such as c-myc, cyclin D1, and cyclin dependent kinase 4 [135]. In 70% of hepatectomized mice, the administration of exogenous IL-22 increased STAT3 activation and the number of 5-bromo-2′-deoxyuridine-positive cells in the liver, pointing to hepatocyte proliferation by IL-22 [136]. A short-term clinical trial of IL-22 reported improved Lille scores and a model for end-stage liver disease scores, implying the severity of liver disease, in patients with AH [137]. Markers of liver regeneration, such as angioprotein-2 and fibroblast growth factor-β, were upregulated, and inflammatory markers were downregulated in these patients [137]. However, in viral hepatitis, IL-22 is considered a pro-inflammatory cytokine and is thought to be involved in the development of liver cancer [138]. Considering that ALD could be aggravated by HCC by the co-contribution of alcohol and viruses, further investigations to prove the safety and effectiveness of IL-22 in ALD are required [139].

### 3.4. Strategies to Relieve Inflammation in ALD

Oxidative stress-induced massive hepatocyte death and failed liver regeneration lead to excessive inflammation [37,40,41]. Inflammation is an important gateway in the pathogenesis of ALD and the progression from mild steatosis to severe liver fibrosis [8,140]. Hence, regulating inflammation remains the most effective treatment strategy for ALD [141,142,143]. Corticosteroids and pentoxifylline, which are currently the recommended therapies for ALD according to clinical guidelines, target inflammation [144,145]. Corticosteroids ameliorate liver inflammation by reducing inflammatory cytokines, such as TNF-α, intercellular adhesion molecule 1, IL-6, and IL-8 [146,147]. Corticosteroid treatment has been shown to alleviate short-term mortality and the incidence of encephalopathy in patients with severe AH [148,149]. Despite the widespread use of corticosteroids, their therapeutic effect is controversial because 40–50% of ALD patients do not respond to treatment [150]. Pentoxifylline is employed as the second-line option for corticosteroid-non-responders and patients with contraindications [151]. Pentoxifylline, a phosphodiesterase inhibitor, suppresses the synthesis of TNF-α by blocking its transcription [152]. Based on findings showing that *Tnf receptor 1* knockout mice exposed to excessive ethanol rarely develop liver injury, TNF-α is considered an inflammatory cytokine in ALD [153]. Several studies have shown that pentoxifylline improves short-term survival in severe acute AH [150,154,155]. However, the number of clinical trials and the number of subjects included in these trials were small, and multiple later trials have failed to provide conclusive evidence of a therapeutic effect of pentoxifylline in ALD [150,156]. Nevertheless, TNF-α is still considered an important therapeutic target in ALD [157,158]. Infliximab is a chimeric mouse/human antibody that binds to TNF-α and blocks its action [159]. A systemic review of the efficacy of infliximab in severe AH found that single dosing of infliximab improves biochemical parameters, such as the amount of total bilirubin, IL-6, IL-8, and neutrophils, and ameliorates the increase in leukocyte infiltration caused by TNF-α, reducing cell injury [160]. However, multiple dosing of infliximab has been shown to be clearly associated with worse outcomes, such as higher mortality and an increased risk of infections [160,161,162,163]. 

The role of IL-1β in controlling inflammation in ALD has also been studied [164,165,166]. In both animal models and patients with ALD, the level of pro-IL-1β significantly increases in liver and serum [167,168]. Inactive pro-IL-1β is cleaved to active IL-1β by caspase-1 in response to inflammatory stimuli [165,169]. Active IL-1β binds to IL-1 receptor type 1 (IL-1R1) and acts in an autocrine or paracrine manner [170]. The IL-1 receptor antagonist (IL-1Ra) competes with IL-1β for IL-1R binding, disrupts IL-1β activity and had been suggested as a therapeutic agent against ALD [167,170]. In *Il-1r1* knockout mice treated with alcohol, the deficiency in IL-1β signaling was associated with a significant decrease in inflammation, damage, and steatosis [167]. Combined treatment with IL-1Ra—known as anakinra—pentoxifylline, and zinc increased the survival rate of AH patients compared with corticosteroid therapy [171]. Based on the findings of a recent clinical trial, interest in the therapeutic potential of IL-1β regulation in ALD has increased [172,173]. In this trial, canakinumab, a recombinant human monoclonal antibody against IL-1β, was shown to directly bind to IL-1β and neutralize its inflammatory activity [172]. As a result, liver histology improved in the canakinumab-treated patients [172]. This recent clinical trial has yet to conduct additional analyses and publish its findings. The findings from these analyses, which are expected in the near future, will reveal the therapeutic potential of canakinumab for severe AH. 

IL-22 regulates inflammation, in addition to promoting liver regeneration [136,174]. In mice exposed to chronic-binge alcohol, IL-22 treatment ameliorated alcoholic liver steatosis, inflammation, and fibrosis, but liver damage was not ameliorated by administration of IL-22 in hepatocyte-specific *Stat3* knockout mice, indicating that the effect of IL-22 in ALD is STAT3 dependent [175]. A clinical trial using F-652, which is a recombinant fusion protein of human IL-22 and immunoglobulin G2 fragment, showed a significant liver improvement with reduced serum levels of pro-inflammatory biomarkers, such as IL-8, MCP-1, and C-reactive protein [137]. However, this trial did not include a placebo group. Randomized placebo-controlled trials are needed to test the efficacy of F-652 in AH [137].

## 4. Stem Cell Therapy for ALD

Despite the promising results of therapeutic candidates for ALD in preclinical studies, their efficacy has been less than expected or not observed in clinical trials. The therapeutic effects of stem cells have been proved in chronic liver diseases such as ALD, NAFLD and acute liver failure [176,177]. Pluripotent stem cells, embryonic stem cells (ESCs) and induced pluripotent stem cells (iPSCs) can differentiate into hepatocyte-like cells [178]. The transplantation of hepatocyte-like cells derived from human ESCs alleviated CCl_4_-induced liver damage by replacing damaged cells and promoting liver regeneration compared to the control group receiving cell medium without ESCs [179]. Transplantation of iPSCs-derived hepatocyte-like cells also improved survival rate of mice with acute liver failure [180]. Although they do not have the ability to differentiate as much as ESCs and iPSCs, MSCs are also multipotent [176,181]. In addition, they have low immunogenicity [176,181]. Hence, MSCs are a commonly used and widely studied in stem cell therapy for liver diseases [176,181]. In high fat diet-induced NAFLD mice, MSC transplantation significantly reduced inflammation and steatosis by suppressing the activation of CD4+ T cells or rescuing mitochondria dysfunction [182,183]. Recently, comprehensive studies have been conducted on the therapeutic effects of EVs on liver diseases [184]. Hence, MSC-based therapies emerge as an attractive treatment option for ALD [21,185,186]. An understanding of the protective effects of MSCs and the underlying mechanisms is needed to develop safe and effective MSC-based therapeutic agents for ALD [185,186,187]. In this section, we review the effects of MSCs and MSC-derived factors on ALD progression (Figure 2).

### 4.1. Direct Transplantation of MSCs in ALD Treatment

Stem cells, including MSCs, have been directly transplanted, and their successful repair effects in various diseases have been proven [187,188,189]. Accumulating evidence has shown the therapeutic functions of MSCs originating from various sources in liver disease, including ALD [22,26,30,190]. Transplantation of BM-derived MSCs has been shown to significantly alleviate alcohol-caused liver damage, such as lipid accumulation, oxidative stress, and inflammation, in mice with AH [191]. Ge et al. [192] showed that BM-MSC transplantation reduced the activation of NK B cells and the secretion of IL-18, a pro-inflammatory cytokine in alcohol-fed mice. Transplantation of human adipose-derived MSCs effectively decreased CYP2E1 expression and increased the activity of the acetaldehyde-metabolizing enzyme ALDH2, alleviating alcohol-induced damage, including lipid accumulation and fibrosis [193]. Based on findings obtained from preclinical studies, MSCs have been administered to patients with alcoholic cirrhosis [194]. In several clinical trials, BM-MSCs administered via intravenous injection significantly improved liver histology and Child–Pugh scores indicating better liver function [195,196,197]. Furthermore, the expression of fibrosis-related markers, such as TGF-β1, collagen type 1, and α-smooth muscle actin, was significantly downregulated in patients after BM-MSC treatment [195,196,197]. However, several hurdles remain to be overcome before BM-MSCs can be approved for clinical applications [198]. Despite the successful outcomes of clinical trials, some researchers have cast doubt on the therapeutic effects of MSC transplantation in patients with alcoholic cirrhosis [199,200]. Rajaram et al. [199] showed that the clinical benefits of MSC transplantation were maintained in patients with advanced cirrhosis for only 8 weeks after MSC transplantation. Spahr et al. [200] reported that BM-MSC transplantation resulted in little improvement in liver histology and function among patients with alcoholic cirrhosis compared with standard medical therapy. In addition, further studies, including large-scale clinical trials, are required to verify the effectiveness of the long-term clinical application of MSCs.

### 4.2. Potential of Cell-Free Strategies for ALD Treatment

MSCs secrete a variety of factors, including cytokines, chemokines, free nucleic acids, and extracellular vesicles (EVs), in response to physiological or pathological stimuli [201]. These MSC-derived secretomes and EVs share many characteristics with their origin, MSCs [202]. They mimic the therapeutic functions of MSCs, including the modulation of immune pathways, cell proliferation, and migration, leading to the creation of a regeneration-favorable microenvironment [202]. The protective role of tumor necrosis factor-inducible gene 6 protein (TSG-6), an anti-inflammatory cytokine released by MSCs, in the liver against NAFLD and fibrosis progression has been proven [203,204,205]. In a recent study, TSG-6 was shown to alleviate the levels of hepatic lipids, MDA and pro-inflammatory cytokines and elevate the amounts of GSH and anti-inflammatory cytokines in TSG-6-treated mice with AH [206]. In this experimental animal model, TSG-6 induced the polarization of Kupffer cells toward an M2 phenotype, and reduced hepatic inflammation and STAT3 activation [206,207]. HGF secreted from skeletal muscle satellite cell-derived MSCs (skMSCs) significantly ameliorated alcohol-induced liver damage in binge alcohol-fed mice [208]. HGF from skMSCs directly recovered the viability and permeability of ethanol-exposed intestinal epithelial cells and intensified the intestinal barrier to suppress hepatic inflammation induced by the leakage of gut-derived hepatotoxins [208]. However, studies on the therapeutic effect of stem cell-derived factors on ALD are limited because no animal models fully mimic the spectrum of human ALD [8,209]. Unlike humans, rodents have a natural aversion to alcohol and a much faster alcohol-catabolizing rate [210]. As a result, alcohol-induced liver pathology is different in rodent models of ALD and patients with ALD [210]. In particular, neutrophil infiltration is hardly detected in rodents during ALD pathogenesis, whereas it is one of the key features of alcoholic steatohepatitis in humans [211].

The therapeutic potential of stem cell-derived factors has been proven in other models of liver disease that share a common pathology with ALD [27,30,212]. These findings point to their therapeutic potential for ALD. For example, human UC-MSC-derived exosomes have been shown to reduce oxidative stress and inhibit apoptosis in mice with CCl_4_-induced liver failure [213]. Furthermore, glutathione peroxidase 1 in EVs derived from human UC-MSCs has been shown to play a key role in the recovery of hepatic oxidant injury and the reversal of oxidative stress-induced apoptosis by inducing extracellular signal-regulated protein kinase 1/2 phosphorylation and Bcl-2 expression [214]. MSC-derived exosomes have been found to improve liver regeneration [215]. EVs released from a human embryonic stem cell line, HuES9-derived MSCs, promoted liver regeneration processes by upregulating priming-phase genes, including proliferating cell nuclear antigen and cyclin D1, in a CCl_4_-induced liver injury model [216]. EVs from placenta-derived MSCs (PD-MSCs) ameliorated hepatic failure caused by bile duct ligation [217]. Furthermore, C-reactive protein in exosomes secreted by PD-MSCs triggered activation of the Wnt signaling pathway and upregulated vascular endothelial growth factor (VEGF) and VEGF receptor 2, which are involved in angiogenesis and liver regeneration [217].

MSC-derived factors have been widely studied in several liver diseases, including NAFLD, acute liver failure, and liver fibrosis, with studies focusing on their immunomodulatory effects [27,218,219]. Numerous studies have revealed that MSC-derived exosomes lower inflammation by reducing inflammatory cytokines or promoting M2 polarization of macrophages [219,220,221]. In a mouse model of acute liver failure, EVs all secreted from either adipose-MSCs or UC-MSCs downregulated inflammatory cytokines, such as IL-6, IL-1β, and TNF-α [222]. The administration of BM-MSC-EVs switched Kupffer cells to an anti-inflammatory phenotype by delivering IL-10 loaded in EVs to target Kupffer cells in mice with hepatic injury induced by hemorrhagic shock [223]. TSG-6 induced the trans-differentiation of activated HSCs into stem-like cells and alleviated liver fibrosis and regenerated hepatic function and structure [205]. Milk fat globule-epidermal growth factor 8 protein inhibited TGF-β signaling by decreasing the expression of TGF-β receptor 1 in HSCs and reducing extracellular matrix deposition and liver fibrosis in CCl_4_-injected mice [224]. In addition, MSC-EVs carrying microRNAs (miR), such as miR-486-5p, miR-150-5p and miR-125b, reduced liver fibrosis by inactivating HSCs [30]. Thus, accumulating data show that the secretome and EVs derived from MSCs mediate therapeutic effects of various types of liver disease, and imply that they have therapeutic potential in ALD treatment. Further investigations are necessary to obtain additional data on characteristics and action mechanism of MSC-released factors to support their therapeutic potential.

## 5. Limitations of Stem Cells Including MSCs Therapy

Because numerous pre-clinical studies have proved the prospective potential of stem cell therapy, an increasing number of clinical trials are being conducted [177,225]. However, actual clinical application of stem cells for chronic liver disease is still challenging [22,177,225]. In order to use stem cells in therapies, exhaustive control for their quality is essential, but characteristics of MSCs are heterogeneous, depending on the health status, genetics, sex, and age of the donor [198,225]. Even if stem cells originate from the same source, it is difficult to maintain the stem cell features, such as stemness, proliferative and migratory capacity, under the diverse culture conditions including confluency, passage number, and culture media supplements [198,225]. Due to these fundamental limitations, criteria for MSC administration including doses, frequencies, and routes of MSC injection have not been established yet. In common with cell-based therapy, similar problems such as the preparation and quality control of MSC-secretomes and -EVs are emerging in cell-free therapy [226]. Currently available techniques for EV isolation are not very efficient because they are time-consuming and have low yields [184,227,228]. In addition, it is hard to distinguish between EVs and natural components such as non-EV proteins and chylomicrons because of a lack of the typical surface markers of EVs [184,228]. Therefore, it is an urgent task to set standards from the preparation to the clinical administration of MSCs and their derived factors to elicit a constant therapeutic effect of MSCs for each patient. In addition, most studies for MSC-derived factors have focused on EVs, not secretory factors, and research on the therapeutic effects of secretory factors from stem cells are in the early stages [181,229]. Therefore, many further studies and verifications are needed to use them for treatment against several diseases including ALD.

## 6. Conclusions

Considering the major impact of alcohol consumption on human culture and society, alcohol can be considered an indispensable factor for humans [230]. Accordingly, alcohol has threatened human health for a long time, with an increasing incidence of ALD [1,5]. Although several drugs are used for the management of ALD, at present, there are no effective drug treatments for ALD [16]. The currently-used drugs serve as supportive therapy to delay the exacerbation of liver disease and liver transplantation timing [16]. Several of these drugs aim to reduce oxidative stress, enhance liver regeneration, and mitigate inflammation by targeting specific steps in the pathogenesis of ALD. MSCs and MSC-released factors, such as the secretome and EVs, are thought to have potential as therapeutic agents for ALD. However, the data obtained from animal models of ALD are limited, and several obstacles to clinical applications of MSCs and MSC-secreted factors remain. Nevertheless, considering their protective effect in various liver diseases, it is clear that they are a promising and attractive resource for developing treatments against ALD. More in-depth investigation is required to solve current challenges associated with the application of MSCs and MSC-derived factors and to support their therapeutic potential in ALD treatment. In conclusion, further comprehensive studies are needed to understand the limitations of current therapy to develop safer and more effective strategies for ALD.

## Figures and Tables

**Figure 1 cells-12-00022-f001:**
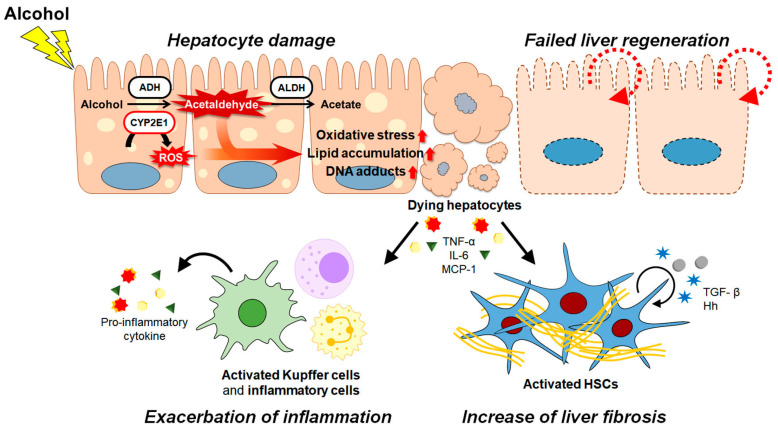
Alcohol metabolism and pathophysiological process in ALD progression. In the liver, alcohol is metabolized to acetaldehyde by alcohol dehydrogenase (ADH) and cytochrome P450 2E1 (CYP2E1). Acetaldehyde is metabolized to acetate by aldehyde dehydrogenase (ALDH). When an excessive amount of alcohol is ingested, CYP2E1 is activated and elevates the levels of acetaldehyde and reactive oxygen species (ROS). Both acetaldehyde and ROS damage the hepatocytes by increasing oxidative stress, lipid accumulation and DNA adducts, leading to hepatocyte death. In addition, alcohol disrupts liver regeneration and aggravates hepatocyte damage. Dying hepatocytes release various cytokines and chemokines, such as tumor necrosis factor-α (TNF-α), interleukin-6 (IL-6), and monocyte chemoattractant protein-1 (MCP-1), which induce activation of inflammatory cells. Activated inflammatory cells produce pro-inflammatory cytokines to exacerbate inflammation. Hepatic stellate cells (HSCs), a major contributor to liver fibrosis, are also activated by cytokines released from dying hepatocytes, and produce extracellular matrix proteins. Activated HSCs release pro-fibrotic factors such as transforming growth factor-β (TGF-β) and Hedgehog (Hh), and maintain their activation status in an autocrine manner or promote activation of inactivated HSCs in a paracrine manner. Eventually, alcohol and its metabolites collectively contribute to the ALD pathogenesis by various pathways and/or mechanisms.

**Figure 2 cells-12-00022-f002:**
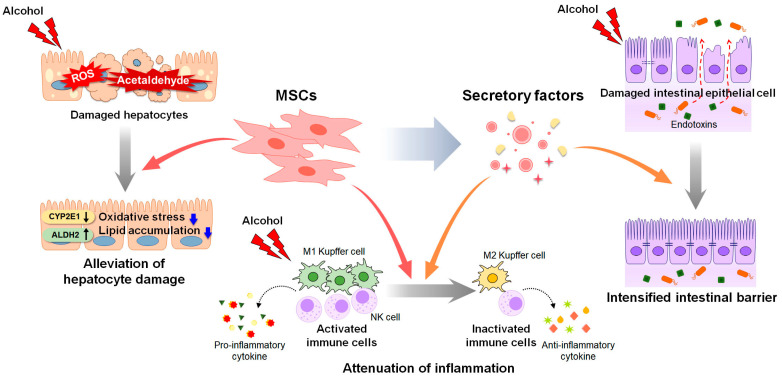
Therapeutic actions of MSCs and MSC-derived secretory factors in ALD. Mesenchymal stem cells (MSCs) and MSC-derived secretome have therapeutic potential for alcohol-induced liver damage. MSC transplantation decreases the expression of cytochrome P450 2E1 (CYP2E1) and increases the activity of the acetaldehyde-metabolizing enzyme aldehyde dehydrogenase 2 (ALDH2), reducing the levels of reactive oxygen species (ROS) and acetaldehyde induced by alcohol. Decreased ROS and acetaldehyde lower oxidative stress and lipid accumulation in hepatocytes. MSCs exert anti-inflammatory property by suppressing the activation of immune cells, including Kupffer cells and natural killer (NK) cells, and the secretion of pro-inflammatory cytokines. MSC-derived secretome such as cytokines, chemokines, free nucleic acids, and extracellular vesicles, alleviates hepatic inflammation by promoting polarization of Kupffer cells toward anti-inflammatory M2 phenotype. In addition, MSC-derived secretome directly improves viability of alcohol-damaged intestinal epithelial cells, and intensifies the intestinal barrier, blocking leakage of endotoxins from intestine.

**Table 1 cells-12-00022-t001:** Summary of therapeutic agents for ALD.

Strategy for Treatment	Therapeutic Agent	Mechanism	Limitations
Blocking alcohol binge	Naltrexone	Blocking opioid receptor to reduce craving for alcohol	Increasing risk of drug-related hepatoxicity Persistent progression of ALD despite of abstinence from alcohol
	Baclofen	Interfering GABA to reduce craving for alcohol	
	Acamprosate	Interfering GABA to reduce craving for alcohol	
	Disulfiram	Inhibiting ADH to cause great hangover after drink	
Lowering oxidative stress	Vitamin E	Neutralizing free radicals, Preventing lipid peroxidation, Inhibiting activation of NF-κB	Insignificant therapeutic effect
	NAC	Increasing glutathione level, Stabilizing glutamate system	Insignificant therapeutic effect in long term study
	SAMe	Regulating glutathione synthesis, Improving mitochondria dysfunction	Insignificant therapeutic effect, Insufficient clinical trials
Promoting liver regeneration	G-CSF	Increasing CD34+ cells, hepatic progenitor cells and HGF level	Unveiled therapeutic mechanism
	IL-22	Promoting cell proliferation	Risk of liver cancer
Alleviating Inflammation	Corticosteroid	Reducing inflammatory cytokine	Insignificant effects to corticosteroid non-responding patients
	Pentoxifylline	Suppressing TNF-α synthesis	Insignificant therapeutic effect
	Infliximab	Binding with TNF-α to block its action Ameliorating leukocyte infiltration	Risk of infection
	IL-1β antagonist (anakinra)	Inhibiting IL-1β by binding with IL-1R1	Insufficient study in ALD
	Canakinumab	Binding with IL-1β to block its action	Insufficient study in ALD
	IL-22, F-652	Activating STAT3	Insufficient clinical trials

GABA, γ-aminobutyric acid; ADH, alcohol dehydrogenase; ALD, alcoholic liver disease; NF-κB, nuclear factor-κB; NAC, N-acetylcysteine; SAMe, S-adenosyl-L-methionine; G-CSF, granulocyte colony stimulating factor; CD34, cluster of differentiation 34; HGF, hepatocyte growth factor; IL-22, interleukin-22; TNF-α, tumor necrosis factor-α; IL-1β, interleukin-1β; IL-1R1, IL-1 receptor type 1; STAT3, signal transducer and activator of transcription 3.

## Data Availability

Not applicable.

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
