# Peer review of "Current Therapeutic Options and Potential of Mesenchymal Stem Cell Therapy for Alcoholic Liver Disease"

_cells, 2022, doi:10.3390/cells12010022_

Round 1
Reviewer 1 Report
In this article, Han et al. have reviewed the potential of mesenchymal Stem Cell Therapy for Alcoholic Liver Disease. ALD is a growing concern worldwide, and new therapies are essential for effectively treating them. Stem cell therapy is undoubtedly an attractive option. The presented article is well written, covering enough background and the available literature about Stem cell therapy for liver disease. However, the authors must address the following concerns to make the article acceptable:
- The authors have concisely explained the pathogenesis of ALD. However, a supportive figure would make understanding easier.
- The applications of Stem cell therapy are well explained. What is lacking is the caveat and challenges of stem cell therapies. This aspect is crucial to keep the article balanced.
- If the literature exists on stem cell therapies (not just MSCs) in liver diseases (not just ALD), highlighting those in a separate section would enhance the applicability of MSCs and increase the completeness of the current article.
Author Response
In this article, Han et al. have reviewed the potential of mesenchymal Stem Cell Therapy for Alcoholic Liver Disease. ALD is a growing concern worldwide, and new therapies are essential for effectively treating them. Stem cell therapy is undoubtedly an attractive option. The presented article is well written, covering enough background and the available literature about Stem cell therapy for liver disease. However, the authors must address the following concerns to make the article acceptable:
1. The authors have concisely explained the pathogenesis of ALD. However, a supportive figure would make understanding easier.
: As you requested, we provided a figure showing ALD pathogenesis. Originally, we did not want to add the figure for ADL pathogenesis because it has been shown in many papers. However, it would be helpful to understand ALD therapy because we reviewed the ALD treatments focusing on the ALD pathogenesis. Thus, we added the figure, as your comment.
2. The applications of Stem cell therapy are well explained. What is lacking is the caveat and challenges of stem cell therapies. This aspect is crucial to keep the article balanced.
: We briefly mentioned the limitation of stem cell therapy. In the revised manuscript, we made a new section for the limitation of stem cell therapy and added more explanation, as your comments. Briefly mentioned limitation of stem cell therapy in a previous version of the manuscript is included in a new section, ‘5 Limitation of stem cells including MSCs therapy’;
“5. Limitations of stem cells including MSCs therapy
Because numerous pre-clinical studies have proved the prospective potential of stem cell therapy, increasing number of clinical trials are being conducted. However, actual clinical application of stem cells for chronic liver disease is still challenging. In order to use stem cells in therapies, exhaustive control for their quality is essential, but characteristics of MSCs are heterogeneous, depending on the health status, genetics, sex, and age of the donor. Even if stem cells originate from the same source, it is difficult to maintain the stem cell features, such as stemness, proliferative and migratory capacity under the diverse culture conditions including confluency, passage number, and culture media supplements. Due to these fundamental limitations, criteria for MSC administration including doses, frequencies, and routes of MSC injection have not been established yet. In common with cell-based therapy, similar problems such as preparation and quality control of MSC-secretomes and -EVs are emerging in cell-free therapy. Currently available techniques for EV isolation are not much efficient because they are time-consuming and have low yields. In addition, it is hard to distinguish between EVs and natural components such as non-EV proteins and chylomicron because of lack of typical surface markers of EVs. Therefore, it is urgent task to set standards from preparation to clinical administration of MSCs and its-derived factors to elicit constant therapeutic effect of MSCs for each patient. In addition, most studies for MSC-derived factors have focused on EVs, not secretory factors, and research on therapeutic effects of secretory factors from stem cells are in early stage. Therefore, much further studies and verifications are needed to use them for treatment against several diseases including ALD.”
3. If the literature exists on stem cell therapies (not just MSCs) in liver diseases (not just ALD), highlighting those in a separate section would enhance the applicability of MSCs and increase the completeness of the current article.
: As you know, stem cell therapy is being studied extensively, and clinical trials using them are in progress in various liver disease including ALD, non-alcoholic fatty liver disease and acute liver failure. However, this review focuses on the current therapeutic options and MScs for ALD. If we add the detailed review for therapeutic role of MSCs in various liver disease, it will be digressed from the main subject of the manuscript. Also, there will be too many contents in one review paper, and too extensive. Hence, we added the short introduction for application of stem cell therapy on various types of liver diseases in Section #4. It seems to be better to introduce the stem cell therapy for ALD; “Therapeutic effects of stem cells have been proved in chronic liver diseases such as ALD, NAFLD and acute liver failure. Pluripotent stem cells, embryonic stem cells (ESCs) and induced pluripotent stem cells (iPSCs), can differentiate into hepatocyte-like cells. Transplantation of hepatocyte-like cells derived from human ESCs alleviated CCl4-induced liver damage by replacing damaged cells and promoting liver regeneration compared to the control group receiving cell medium without ESCs. Transplantation of iPSCs-derived hepatocyte-like cells also improved survival rate of mice with acute liver failure. Although they do not have the ability to differentiate as much as ESCs and iPSCs, MSCs are also multipotent. In addition, they have low immunogenicity. Hence, MSCs are a commonly used and widely studied in stem cell therapy for liver diseases. In high fat diet-induced NAFLD mice, MSC transplantation significantly reduced inflammation and steatosis by suppressing activation of CD4+ T cells or rescuing mitochondria dysfunction. Recently, comprehensive studies have been conducted the therapeutic effects of EVs on liver diseases.”
Reviewer 2 Report
This review provides current or possible treatments for alcoholic liver disease and contributed to the understanding of the development of effective and safe treatments for alcoholic liver disease. The manuscript is rich in content and well-organized. However, I have some concerns.
1. The topic is "Current Therapeutic Options and Potential of Mesenchymal Stem Cell Therapy for Alcoholic Liver Disease", but there is not that much content about stem cell therapy. So I think the theme or the content should be adjusted.
2. The literature needs to be updated, especially articles from the last five years.
Author Response
This review provides current or possible treatments for alcoholic liver disease and contributed to the understanding of the development of effective and safe treatments for alcoholic liver disease. The manuscript is rich in content and well-organized. However, I have some concerns.
1. The topic is "Current Therapeutic Options and Potential of Mesenchymal Stem Cell Therapy for Alcoholic Liver Disease", but there is not that much content about stem cell therapy. So I think the theme or the content should be adjusted.
: We appreciate your comments. As we mentioned on the manuscript, stem cell therapy for ALD has not been widely studied compared to other liver disease because of various limitations such as absence of animal models mimicking ALD pathology of human patients and lack of clinical studies with large cohorts. However, it is clear that stem cell therapy is a promising therapeutic strategy for ALD, considering its therapeutic effect on other liver disease. Hence, to add more about stem cell, we briefly described MSCs and other types of stem cell therapy in other liver diseases and added limitation of stem cell therapy in the revised manuscript. Please see the answer to reviewer #1’s comments.
2. The literature needs to be updated, especially articles from the last five years.
: As you requested, we added more recent studies and updated references. This is a list of added references; [Stem Cells Transl Med 2022, 11, 900-911./ Stem Cell Res Ther 2022, 13, 356./ Signal Transduct Target Ther 2022, 7, 92./ Stem Cell Res Ther 2021, 12, 602./ Mol Med Rep 2018, 17, 3769-3774./ Stem Cell Res Ther 2021, 12, 521./ Am J Transl Res 2021, 13, 3954-3966./ Stem Cells Int 2021, 2021, 2616807./ J Extracell Vesicles 2021, 10, e12128./ Cells 2019, 8./ Cell Death Dis 2022, 13, 580.]
Round 2
Reviewer 1 Report
The authors have addressed my comments satisfactorily.
Reviewer 2 Report
I have no comments now.